# High Endemicity of Soil-Transmitted Helminths in a Population Frequently Exposed to Albendazole but No Evidence of Antiparasitic Resistance

**DOI:** 10.3390/tropicalmed4020073

**Published:** 2019-04-27

**Authors:** Gabriela Matamoros, María Mercedes Rueda, Carol Rodríguez, Jose A. Gabrie, Maritza Canales, Gustavo Fontecha, Ana Sanchez

**Affiliations:** 1Department of Health Sciences, Brock University, St. Catharines, ON L2S 3A1, Canada; jgabrie@brocku.ca; 2Instituto de Investigaciones en Microbiología, Universidad Nacional Autónoma de Honduras, Tegucigalpa, Honduras; maria.rueda@unah.edu.hn (M.M.R.); carol.rodriguez@unah.edu.hn (C.R.); maritza.canales@unah.edu.hn (M.C.); gustavo.fontecha@unah.edu.hn (G.F.)

**Keywords:** soil-transmitted helminths, anthelminthic resistance, benzimidazole, single nucleotide polymorphisms, mass drug administration, Honduras

## Abstract

Introduction: Soil-transmitted helminths (STHs) are gastrointestinal parasites widely distributed in tropical and subtropical areas. Mass drug administration (MDA) of benzimidazoles (BZ) is the most recommended for STH control. These drugs have demonstrated limited efficacy against *Trichuris trichiura* and the long-term use of single-dose BZ has raised concerns of the possible emergence of genetic resistance. The objective of this investigation was to determine whether genetic mutations associated with BZ resistance were present in STH species circulating in an endemic region of Honduras. Methods: A parasitological survey was performed as part of this study, the Kato–Katz technique was used to determine STH prevalence in children of La Hicaca, Honduras. A subgroup of children received anthelminthic treatment in order to recover adult parasite specimens that were analyzed through molecular biology techniques. Genetic regions containing codons 200, 198, and 167 of the β-tubulin gene of *Ascaris lumbricoides* and *Trichuris trichiura* were amplified and sequenced. Results: Stool samples were collected from 106 children. The overall STH prevalence was 75.47%, whereby *T. trichiura* was the most prevalent helminth (56.6%), followed by *A. lumbricoides* (17%), and hookworms (1.9%). Eighty-five sequences were generated for adjacent regions to codons 167, 198, and 200 of the β-tubulin gene of *T. trichiura* and *A. lumbricoides* specimens. The three codons of interest were found to be monomorphic in all the specimens. Conclusion: Although the inability to find single-nucleotide polymorphisms (SNPs) in the small sample analyzed for the present report does not exclude the possibility of their occurrence, these results suggest that, at present, Honduras’s challenges in STH control may not be related to drug resistance but to environmental conditions and/or host factors permitting reinfections.

## 1. Introduction

Soil-transmitted helminths (STHs) are gastrointestinal parasites widely distributed in tropical and subtropical areas, with an elevated prevalence in regions with a high poverty index [1].

The main STH species are *Trichuris trichiura*, *Ascaris lumbricoides*, and hookworms, mainly *Necator americanus* in the Americas. Infections caused by these parasites are transmitted by ingestion or contact with fecally contaminated soil. Children are more at risk of acquiring these infections which, due to their insidious nature, are associated with detrimental consequences, primarily growth and cognitive impairment [2].

The recommended control strategy for STH infections is the administration of periodic treatment of at-risk populations living in endemic regions [3]. Anthelminthics from the benzimidazole (BZ) family—albendazole (ABZ) (400 mg) and mebendazole (MBZ) (500 mg)—are the most recommended pharmaceuticals, due to their low cost and relative effectiveness when administered in single-dose regimes [2,3].

Benzimidazoles’ main mode of action consists of binding to the β-tubulin molecule, inhibiting the polymerization of microtubules. This results in alteration of both cellular structure and function, leading to parasites’ death [4,5]. Although BZ are convenient to use during mass drug administration (MDA) programs, their limited efficacy against *T. trichiura* infections has been widely demonstrated [6,7,8,9,10]. Thus, *T. trichiura* has become the most prevalent STH in Latin America [11,12,13].

Although this limited efficacy may be due to several factors, concerns have been raised that genetic resistance to BZ may be involved in the prevalence of *T. trichiura*. These concerns are founded on unambiguous biological evidence and clinical experience from the field of veterinary medicine, [14,15,16] where three amino acid substitutions in codons 167, 198, and 200 of the β-tubulin gene of *Haemonchus contortus* have been associated with BZ resistance [17,18,19].

Several molecular methods, such as qPCR, deep amplicon sequencing, pyrosequencing, and Sanger sequencing [20,21,22,23], have been used to determine the presence of the single-nucleotide polymorphisms (SNPs) associated with BZ resistance. Multiple studies have demonstrated the existence of SNPs in codons 167, 198, and 200 in *T. trichiura*; in addition, mutations in codon 167 and 198 have been found in *A. lumbricoides,* and SNPs in codons 198 and 200 have been detected in hookworms recovered from humans [7,20,24,25,26]. Multiple studies have also reported these SNPs occurring in other gastrointestinal nematodes of veterinary importance [18,27,28,29,30].

The emergence of anthelminthic resistance in gastrointestinal nematodes infecting animals raises concerns regarding the possible development of antimicrobial resistance in helminth species infecting humans. This is especially true in regions where MDA programs exert selective pressure on the parasites, potentially favoring parasites with naturally occurring mutated genotypes [8,31]. Following the recommendations of the World Health Organization (WHO), Honduras is an STH endemic country aiming to reach a 75% coverage of all children at risk [2]. To this effect, national deworming activities with single-dose ABZ (400 mg) tablets are implemented twice a year, around April and November. Despite these efforts, STHs remain highly endemic in certain rural communities [12,32,33,34].

The objective of this investigation was to determine the presence of genetic mutations associated with anthelminthic resistance in a highly endemic community frequently exposed to albendazole. Specifically, we aimed at determining the frequency of the F200Y, E198A, and F167Y SNPs in parasites recovered from children living in a highly endemic region in Honduras.

## 2. Material and Methods

This study was nested within an STH cross-sectional parasitological survey undertaken in the village of La Hicaca and nearby hamlets, located in a rural mountainous area in the department of Yoro, northern Honduras. The study took place in June 2016.

For the parasitological survey, children were invited to participate in the study through their parents. After obtaining parental informed consent and children’s oral assent, participants were provided with a stool sample collection kit. Stool samples were analyzed using the Kato–Katz technique to determine the presence and intensity of infections. In order to identify hookworm infections, smears were read under the microscope within 30–45 min after preparation. The latter is estimated following WHO recommendations that consider the number of eggs per gram (epg) of stools counted in the Kato–Katz as follows: For *A. lumbricoides*, 1–4999 epg (light); 5000–49,999 epg (moderate); and >50,000 epg (heavy). For *T. trichiura*, 1–999 epg (light); 1000–9999 epg (moderate); and >10,000 epg (heavy). For hookworms, 1–1999 epg (light); 2000–3999 epg (moderate); and >4000 epg (heavy) [35].

A subgroup of eight children harboring infections of heavy and moderate intensity were invited to receive a special deworming treatment in order to recover adult parasite specimens (Table 1). The remaining infected children were treated with a single-dose of 400 mg ABZ. Treatments were administered by the health center’s registered nurse.

Children in the special treatment subgroup were asked to bring 24 h stool samples to the health center over four consecutive days. Those samples were exhaustively examined for the presence of adult worms. The fecal matter was filtered through sieves with pores of 2 mm in diameter. All collected worms were washed with distilled water and saline solution (0.85% (*w*/*v*) NaCl) and stored individually in 70% ethanol at room temperature for further molecular analysis.

DNA was extracted through the Automate Express™ system using the commercial kit PrepFiler Express BTA™, according to the manufacturer’s protocol. Specific primers were used in order to amplify the segments surrounding the codons 200, 198, and 167 in the β-tubulin gene of both parasites. *T. trichiura* codons 200, 198, and 167 were amplified using a single PCR utilizing the primers previously described by Hansen et al. [7]. The PCR mix was prepared using 12.5 µL of 2× Promega™ PCR master mix, 1 µL of forward and reverse primers at 10 µM concentration each, 1 µL of bovine serum albumin (BSA) (20 mg/mL), and 2 µL of template DNA at a minimum concentration of 20 ng/mL to obtain a final volume of 25 µL with nuclease-free water. The cycling conditions for this PCR were as follows: 5 min at 95 °C; followed by 35 cycles of 95 °C for 1 min, 63 °C for 1 min, and 72 °C for 1.5 min; and a final extension of 72 °C for 5 min. The amplification of the regions around codons 200 and 198 in *A. lumbricoides* were performed through a semi-nested PCR approach with one forward primer (5′-AGAGCCACAGTTGGTTTAGATACG-3′) and two reverse primers (PCR 1: 5′-AGGGTCCTGAAGCAGATGTC-3′; PCR 2: 5′-CAGATGTCGTACAAAGCCTCATT-3′) following the previously mentioned PCR conditions, except for the annealing temperature at 64 °C in the semi-nested PCR [25]. On the other hand, amplification of codon 167 was performed through a single PCR which contained 25 µL of 2× Promega™ master mix, 2 µL of forward (5′-CCGTGAAGAATACCCCGAC-3′) and reverse primer (5′- GATGAACGGACAACGTTGC-3′), 2 µL of BSA (20 mg/mL), and 4 µL of template DNA to obtain a final volume of 50 µL with nuclease-free water, following amplification conditions described by Diawara et al. [21]. Amplicons were sequenced by Macrogen Corp. at Maryland, USA, using the same primers as in the PCR. Sequences were aligned in the Geneious ™ software, R8.1 version, using MUSCLE alignment.

The point prevalences of STH infections based on Kato–Katz results were calculated with 95% confidence intervals (CI). Frequencies of SNPs in the β-tubulin gene in both parasites were reported as percentages.

## 3. Results

### 3.1. STH Prevalence in Studied Population

A total of 106 stool samples were collected from children whose ages ranged from 0.6 to 13 years (mean age = 6.3, SD = 3.1). The overall STH prevalence was 61.32% (CI: 95%, 52.05%, 70.59%). *T. trichiura* was the most prevalent helminth, with a point prevalence of 56.6% (CI: 95%, 47.17%, 66.04%) followed by *A. lumbricoides* with 17.0% (CI: 95%, 9.83%, 24.13%), and hookworms 2% (CI: 95%, 0.00%, 4.48%). Table 2 shows the frequency of each infection, organized by single or multiple helminth species.

Table 3 shows percentages of infection intensity per parasite. In general, 77% of all infections were light, 22% were of moderate intensity, and only 1.7% were severe. The only parasite causing severe infections was *T. trichiura.*

### 3.2. Genotyping of Codons 200, 198, and 167 from the β-Tubulin Gene in T. trichiura and A. lumbricoides

Out of 452 adult worms collected from 8 children, 123 specimens were genotyped and 85 sequences (40 *A. lumbricoides* and 45 *T. trichiura*) with a length of 600 bp from the β-tubulin gene fragment were obtained. Three codons (167, 198, and 200) within the β-tubulin gene were genotyped with the purpose of identifying the substitutions associated with BZ resistance (F200Y, E167A, and F198Y). Figure 1 shows the electropherograms of analyzed samples, all of them demonstrating the wild type (WT) genotype. The three codons of interest were found to be monomorphic. SNPs associated with BZ resistance were not identified among the specimens analyzed.

## 4. Discussion

The STH prevalence observed in the study population was 75.47% which, according to WHO’s guidelines, is considered highly endemic [35,36]. A study in the same community in the previous year by Sanchez et al. reported an overall STH prevalence of 69% in a sample of 130 children [13]. This continued high endemicity demonstrates that in the absence of safe water, sanitation, and hygiene, STH transmission continues unabated and reinfections are the norm. Despite MDA programs across the country, there are many hyperendemic foci where STH prevalence remains not only an important public health problem, but a potential medical problem as well [13,32,33,34,37].

On a more positive note, our study shows that infection intensities have shifted and are now mostly light to moderate, and that heavy intensity infections are becoming less frequent [12,13,34,37]. According to the WHO, such is the main goal of MDA programs—decreasing parasitic burden and associated morbidity even when the interruption of transmission is not fully accomplished [2].

In terms of individual STH species, our results coincide with previous Honduras-based studies reporting *T. trichiura* as the most prevalent STH, followed by *A. lumbricoides* and hookworms [12,37].

Our findings also revealed that *T. trichiura* was the only parasite causing heavy infections. Persistent *T. trichiura* prevalence and heavy worm burden in a population that regularly receives anthelminthic treatment with BZ [13,38] may be indicative of parasitic resistance associated with SNPs in the β-tubulin gene [25,39].

The present study analyzed 85 sequences of STH adult specimens recovered from infected children in La Hicaca, Honduras. Forty *A. lumbricoides* and 45 *T. trichiura* sequences were generated and three codons were analyzed. We did not identify the SNPs associated with BZ resistance in this sample. Our results agree with previous studies in African and Asian countries that have not been able to identify these SNPs in specimens recovered from humans and other animals [7,24]. A study conducted in several regions of Brazil did not identify SNPs associated with benzimdazole resistance in codons 198 and 167 of DNA extracted from *A. lumbricoides* eggs [26]. By contrast, studies undertaken in Haiti, Kenya, and Panama have found elevated frequencies of these SNPs, especially in codon 200 of the β-tubulin gene in *T. trichiura* after anthelminthic treatment administration [20,25]. In the case of *A. lumbricoides,* high frequency mutations have only been identified in codon 167 of the β-tubulin gene [25].

Based on the known emergence of resistance against multiple anthelminthic drugs among veterinary parasites [30,40] and the published data that demonstrate the emergence of SNPs associated with BZ resistance in parasites infecting humans [20,25], active surveillance for drug resistance is necessary in territories where MDAs have been implemented. Further, implementing combination therapies would reduce the probability of the emergence of genetic resistance, while also resulting in the accelerated impact of STH drug therapy [41].

## 5. Conclusions

The inability to find SNPs associated with BZ resistance in the small sample of *T. trichiura* and *A. lumbricoides* adult specimens analyzed for the present report does not exclude the possibility of their occurrence. Another possibility, as recently suggested, is that other unidentified resistance mechanisms are playing a role in contributing to a high prevalence of these helminths [42]. At present, however, Honduras’s challenges in STH control may not be related to drug resistance but to low drug efficacy, environmental conditions, and/or host factors permitting reinfections.

## Figures and Tables

**Figure 1 tropicalmed-04-00073-f001:**
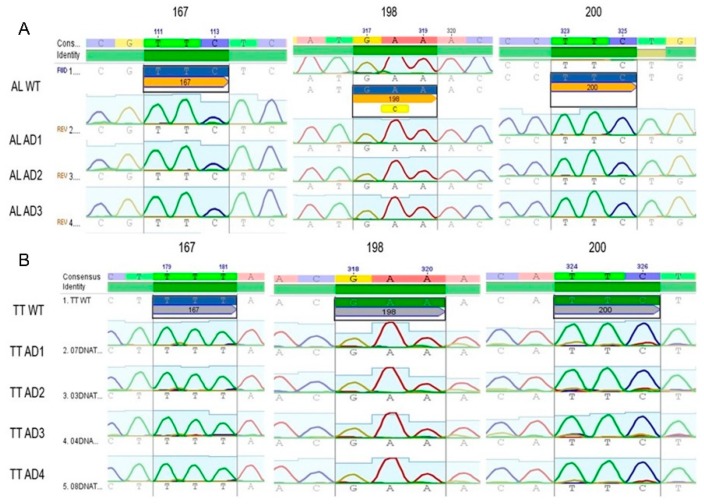
Electropherogram of sequences obtained from adult *Trichuris trichiura* and *Ascaris lumbricoides* specimens recovered from Honduran SAC and Pre-SAC. Subsection A illustrates three sequences (ALAD1, ALAD2, and ALAD3) from adult *Ascaris* specimens and subsection B shows 4 sequences (TTAD1, 2, 3, and 4) compared to a consensus sequence (*A. lumbricoides* GenBank accession. number: FJ501301.1 and for *T. trichiura*, AF034219.1), the codons of interest are represented by their corresponding number, which shows that there are no polymorphisms in any of the samples shown.

**Table 1 tropicalmed-04-00073-t001:** Treatment scheme used to recover adult worms from infected children, according to type of infection. ABZ: albendazole.

Infection	Treatment per Day			
	DAY 1	DAY 2	DAY 3	DAY 4
*Trichuris trichiura*	Pyrantel-oxantel ^A^	Pyrantel-oxantel	Pyrantel-oxantel	ABZ
*Ascaris lumbricoides*	Piperazine ^B^	ABZ ^C^	ABZ	ABZ
Mixed	Pyrantel-oxantel and piperazine	Pyrantel-oxantel	Pyrantel-oxantel	ABZ

^A^ dose of 10 mg/kg, ^B^ dose of 75 mg/kg, and ^C^ dose of 400 mg.

**Table 2 tropicalmed-04-00073-t002:** Prevalence of soil-transmitted infections diagnosed with Kato–Katz in preschool-aged children (Pre-SAC) and school-aged children (SAC) in La Hicaca, Yoro, Honduras (*n* = 106).

Infection	Pre-SAC		SAC		TOTAL	
	(0–6 Years)		7–13 Years			
	*n*	%/106	*n*	%/106	*N*	%/106
*T. trichiura* only	28	26.4%	18	17%	46	43.4%
*T. trichiura + A. lumbricoides*	5	4.7%	7	6.6%	12	11.3%
*A. lumbricoides* only	2	1.9%	3	2.8%	5	4.7%
Hookworm only	0	0%	0	0%	0	0%
Hookworm *+ T. trichiura*	1	1%	0	0	1	1%
Hookworm *+ A. lumbricoides + T. trichiura*	1	1%	0	0	1	1%
No helminths observed	23	21.7%	18	17%	41	38.7%
Total	60	56%	46	43.4%	106	100%

**Table 3 tropicalmed-04-00073-t003:** Cases with light, moderate or heavy soil-transmitted helminth (STH) infections among infected children (*n* = 65).

Infection	Cases of Intensity of Infection *		
	Light	Moderate	Heavy
*T. trichiura*	48/60 (80%)	11/60 (18%)	1/60 (2%)
*A. lumbricoides*	12/18 (67%)	6/18 (33%)	0/18 (0%)
Hookworm	2/2 (100%)	0/0 (0%)	0/0 (0%)

* Cases include single and multiple helminth species infections.

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
