# Peer review of "High Endemicity of Soil-Transmitted Helminths in a Population Frequently Exposed to Albendazole but No Evidence of Antiparasitic Resistance"

_tropicalmed, 2019, doi:10.3390/tropicalmed4020073_

Round 1
Reviewer 1 Report
1. Table 2: Are all these single infections? Since mixed infection is likely to occur in endemic areas. On the other hand, if the authors intentionally chose single-infection samples only then this selection should be mentioned in the method section.
2. Regarding the worms recovered from the collected stool after treatment, is the species corresponding to that found in Kato-Katz method before treatment?
3. Do the authors have information regarding the history of MDA treatment (how long and the frequency) in the areas where the samples came from?
4. Results, line 137: the 452 worms were recovered from how many individuals? At the end, the 85 sequences were from how many individuals?
5. Table 3: besides the percentages, please show the number of cases per infection intensity by species
6. Figure 1 legend: Subsection B should be 4 sequences (TTAD1, 2, 3, 4)
Author Response
Comments and Suggestions for Authors and Author's Reply to the Review #1 Report
1. Table 2: Are all these single infections? Since mixed infection is likely to occur in endemic areas. On the other hand, if the authors intentionally chose single-infection samples only then this selection should be mentioned in the method section.Thank you for this observation. Indeed, the table was not clear so we have now included all data with single and multiple infections and corresponding percentages.
2. Regarding the worms recovered from the collected stool after treatment, is the species corresponding to that found in Kato-Katz method before treatment?
Yes, the helminths recovered corresponded to the Kato-Katz findings.
3. Do the authors have information regarding the history of MDA treatment (how long and the frequency) in the areas where the samples came from?
Yes, this information is included in the manuscript. On line 69-71 we had stated “To this effect national deworming activities with single dose 400 mg albendazole tablets are implemented twice a year, around April and November. Despite these efforts STH remain highly endemic in certain rural communities”
4. Results, line 137: the 452 worms were recovered from how many individuals? At the end, the 85 sequences were from how many individuals?
On line 83 (in Methods section) and line 137 (results section), we have added the number of children (i.e., eight) from whom the worms were recovered.
5. Table 3: besides the percentages, please show the number of cases per infection intensity by species
We have modified Table 3 to adopt this recommendation.
6. Figure 1 legend: Subsection B should be 4 sequences (TTAD1, 2, 3, 4)
The reviewer is correct. The legend has been modified accordingly.
Reviewer 2 Report
Abstract
Minor, and not sure what the journal requirements are, but new paragraph for methods, results and conclusions would be nice, and perhaps bold those headings.
Line 49: BZ was used as an abbreviation in the abstract, to keep consistent I would define at first mention (line 47) then use the abbreviation thereafter
Line 100: italics Trichuris trichiyra
Line 102: ? mark after each
Line 107: italics A. lumbricoides
Table 2: Mention in title that prev is by KK, and also somewhere in the text line 121-126
Table 3: Either make all numbers decimals, or round up heavy to 2. Leaves 0.3% of infections as unknown intensity. I would also prefer this table, or somewhere in the text, to indicate what constitutes light, heavy, and moderate intensity infections.
KK is not always the best method for hookworm as the eggs lyse fairly quickly after deposition thus slides need to made quickly and read quickly. What was the time frame between collection and making the slides, and then viewing them?
Line 147: this bit needs to be clearer, states three sequences but then in the brackets are 4 sequence labels? Make clear which are study sequences of Trichuris. The line for Ascaris above is good and clear.
Line 170: BZ used here, and benzimdiazole used in line 173. Keep it consistent.
Author Response
Reviewer 2
Comments and Suggestions for Authors and Author's Reply to the Review #2 Report
Abstract
Minor, and not sure what the journal requirements are, but new paragraph for methods, results and conclusions would be nice, and perhaps bold those headings.
The journal style will determine the layout of the abstract
Line 49: BZ was used as an abbreviation in the abstract, to keep consistent I would define at first mention (line 47) then use the abbreviation thereafter
This is a correct observation: in line 43 we had already defined the abbreviation: “Anthelminthics from the benzimidazole (BZ) family —albendazole (ABZ)…” however, we were not using it afterwards. We have now made the suggested change for consistency. We only left the full word at the beginning of the sentence in line 49.
Line 100: italics Trichuris trichiura
We have made the change. Please note that because we added some text, the line has moved to line 106.
Line 102: ? mark after each
We have removed the question mark. Please note that because we added some text, the line has moved to line 109.
Line 107: italics A. lumbricoides
We have made the change. Please note that because we added some text, the line has moved to line 113.
Table 2: Mention in title that prev is by KK, and also somewhere in the text line 121-126
The title of this table has changed to
“Prevalence of soil-transmitted infections diagnosed with Kato-Katz in pre-school aged children (Pre-SAC) and school-aged children (SAC) in La Hicaca, Yoro, Honduras (N=106)”included a sentence to that extent on Line 124.
Table 3: Either make all numbers decimals, or round up heavy to 2. Leaves 0.3% of infections as unknown intensity.
We have modified the table to follow this recommendation.
I would also prefer this table, or somewhere in the text, to indicate what constitutes light, heavy, and moderate intensity infections.
We have included this information in the methodology section (lines 83-87) and cited the WHO (2012) accordingly.
KK is not always the best method for hookworm as the eggs lyse fairly quickly after deposition thus slides need to made quickly and read quickly. What was the time frame between collection and making the slides, and then viewing them?
a) This is absolutely correct. Due to the high ambient temperature in the locality (about 35-40 Celsius), we read the K-K smears within 30-45 minutes after preparation. This was inserted in lines 84-85.
b) We have actually published about the hookworm egg fragility. See here: http://www.bvs.hn/RMH/pdf/2012/pdf/Vol80-3-2012-4.pdf
Line 147: this bit needs to be clearer, states three sequences but then in the brackets are 4 sequence labels? Make clear which are study sequences of Trichuris. The line for Ascaris above is good and clear.
Yes, very good observation. We’ve made the correction.
Line 170: BZ used here, and benzimdiazole used in line 173. Keep it consistent.
We have used the BZ abbreviation consistently.